# Intraspecific Diversity and Pathogenicity of *Bacillus thuringiensis* Isolates from an Emetic Illness

**DOI:** 10.3390/toxins15020089

**Published:** 2023-01-18

**Authors:** Jintana Pheepakpraw, Thida Kaewkod, Maytiya Konkit, Sasiprapa Krongdang, Kanyaluck Jantakee, Rueankaew Praphruet, Sakunnee Bovonsombut, Aussara Panya, Yingmanee Tragoolpua, Niall A. Logan, Thararat Chitov

**Affiliations:** 1Department of Biology, Faculty of Science, Chiang Mai University, Chiang Mai 50200, Thailand; 2Doctor of Philosophy Program in Applied Microbiology (International Program), Faculty of Science, Chiang Mai University, Chiang Mai 50200, Thailand; 3Division of Microbiology, Faculty of Science and Technology, Nakhon-Pathom Rajabhat University, Nakhon-Pathom 73000, Thailand; 4Faculty of Science and Social Sciences, Burapha University, Sa Kaeo Campus, Sa Kaeo 27160, Thailand; 5Institute of Product Quality and Standardization, Maejo University, Chiang Mai 50290, Thailand; 6Environmental Science Research Center (ESRC), Faculty of Science, Chiang Mai University, Chiang Mai 50200, Thailand; 7Formerly of Glasgow Caledonian University, Glasgow G4 0BA, UK

**Keywords:** foodborne intoxication, foodborne outbreak, emetic toxin, *Bacillus thuringiensis*, *Bacillus cereus* group, *cereulide synthetase* gene, cytotoxicity, Caco-2 cells

## Abstract

This study describes an emetic food-borne intoxication associated with a *Bacillus cereus* group species and the characterization of the bacterial isolates from the incident in aspects of molecular tying, genetic factors, cytotoxicity, and pathogenic mechanisms relating to emetic illness. Through the polyphasic identification approach, all seven isolates obtained from food and clinical samples were identified as *Bacillus thuringiensis*. According to multilocus sequence typing (MLST) analysis, intraspecific diversity was found within the *B. thuringiensis* isolates. Four allelic profiles were found, including two previously known STs (ST8 and ST15) and two new STs (ST2804 and ST2805). All isolates harbored gene fragments located in the *cereulide synthetase* (*ces*) gene cluster. The heat-treated culture supernatants of three emetic *B. thuringiensis* isolates, FC2, FC7, and FC8, caused vacuolation and exhibited toxicity to Caco-2 cells, with CC_50_ values of 56.57, 72.17, and 79.94 µg/mL, respectively. The flow cytometry with the Annexin V/PI assay revealed both apoptosis and necrosis mechanisms, but necrosis was the prominent mechanism that caused Caco-2 cell destruction by FC2, the most toxic isolate.

## 1. Introduction

*Bacillus cereus* has long been recognized as a food-borne pathogen, one capable of producing two distinct types of toxins: diarrheagenic and emetic toxins [1,2]. Some other species that are closely related to *B. cereus*, known as the *B. cereus* group [2], were also occasionally found involved in food-borne outbreaks, such as *B. thuringiensis* and, more rarely, *B. mycoides* [3]. It is possible that some of the *B. cereus* group species might have been misidentified as *B. cereus* at the time of the outbreaks [3] or that *B. cereus* group species other than *B. cereus* have become emerging food-borne pathogens [4]. In addition, the potential of the *B. cereus* group species to cause food-borne illnesses has been increasingly recognized in recent years [5].

*Bacillus thuringiensis*, a species closely related to *B. cereus*, is better known for its ability to produce insecticidal toxins. *B. thuringiensis* has been widely used as a biocontrol agent [6,7,8,9] and its toxin genes have been transferred to genetically modified crops [10,11]. Members of the *B. thuringiensis* species have occasionally been found in food [12,13,14]. Enterotoxin genes, commonly present in *B. cereus*, are also widely distributed among *B. thuringiensis* isolates [5,15,16]. This species has been implicated in food-borne illnesses, but most of the *Bacillus* food-borne outbreak cases involved *B. cereus* [3,17]. There has been some indication of the presence of pathogenicity factors in some *B. thuringiensis* outbreak strains. In the epidemiology study by Bonis et al. in 2021 [18], *B. thuringiensis* strains retrieved from food-borne outbreaks in France during 2007–2017 were collected, and their phenotypic, genotypic, and genome characteristics were investigated. Virulence factors, including lecithinase activity, hemolysis activity, and virulence genes (*cytK2*, *nheABC*, and *hblCDA*), were found to be widely distributed (97–100%) in the outbreak strains and could be associated with the gastrointestinal symptoms in the outbreak history records. These virulent factors are mostly related to the diarrheal type of illness [18]. Little is known about this species’ association with emetic illness and the pathogenicity factors associated with it.

Following an incident of emetic food-borne outbreak in Glasgow, UK, *Bacillus thuringiensis* was isolated from food and vomit samples. The incident occurred in December 2007 and involved two female children, ages three and four, from one family. The meal was prepared using fresh pasta, smoked mackerel, and mussels in a garlic butter sauce, all purchased from a supermarket. Within a few hours of purchase, the pasta was cooked in boiling water, the smoked mackerel was combined with the mussels in a garlic butter sauce, and the sauce portion was reheated. The meal was served immediately after preparation. Approximately 4 h after consumption, the older child started vomiting, and several further episodes of vomiting occurred over the next 3 h. The younger child vomited only once, about 5 h after the consumption of the meal. There were no other symptoms. The children had eaten similar-sized portions, but their parents, who had eaten larger portions of the same food, did not show any symptoms of illness.

Specimens of the meal leftovers (the sauce portion) and vomit from one child were examined for microbiological content and possible toxin residue. The level of presumptive *B. cereus* group colonies was recorded as less than 1.0 × 10^2^ cfu/g in food and 8.0 × 10^2^ cfu/g in the vomit sample. These strains were subsequently identified as *B. thuringiensis*. Neither coagulase-positive staphylococci nor staphylococcal enterotoxins were found in the samples.

This paper describes further characterization of the *B. thuringiensis* isolates involved in the emetic outbreak and their heat-stable toxin. It particularly focuses on their biochemical characteristics, diversity, genetic factors associated with emetic toxin production, and cytotoxicity.

## 2. Results and Discussion

### 2.1. Morphological and Biochemical Characterization of Food-Borne Outbreak Isolates

All the *B. thuringiensis* isolates from food and the vomit samples were Gram-positive, spore-forming rods with unswollen sporangia, bearing roughly bipyramidal toxin crystals (Figure 1). They could grow at 45 °C and below, but not at 48 °C or 50 °C. The API profiles of the isolates varied, but they all showed the typical biochemical reaction patterns for *B. cereus*/*B. thuringiensis* species (Appendix A). All isolates were able to degrade starch and utilize salicin.

Cases of *B. cereus* emetic illness are usually associated with rice and pasta dishes [18,19,20,21,22,23], but the organisms involved in this small food-borne outbreak were isolated from the mackerel and mussels in the garlic butter sauce, not from the pasta. However, an extensive study by Messelhäusser et al. (2014) [24] showed that emetic toxin-producing strains belonging to the *B. cereus* group were not only found in foods with high carbohydrate content, such as pasta or rice, as was earlier found. These observations have implications for the breadth of sampling that should be carried out in the investigation of emetic outbreaks. The isolates had biochemical characteristics typical of *B. cereus/B. thuringiensis*, but it was interesting to note that the present isolates could degrade starch and utilize salicin, unlike many emetic *B. cereus* strains previously characterized [25,26,27]. This recalls the comment made by Thorsen et al. (2006) [28] that not all emetic toxin-producing *B. cereus* strains are unable to hydrolyze starch or produce acid from salicin.

### 2.2. Molecular Identification and Typing of Isolates from Emetic Illness

#### 2.2.1. Identification of Isolates Using *16S rRNA* Gene Sequence Analysis

The *16S rRNA* sequences of the *B. thuringiensis* isolates, two from food (FC1 and FC2; obtained from mussels and smoked mackerel in garlic butter sauce), and five from the vomit sample of one patient (FC6, FC7, FC8, FC9, and FC10), were determined. The *16S rRNA* Blast results against strains in the NCBI database confirmed their identity as *B. thuringiensis* (Table 1).

The sequences of the *B. thuringiensis* isolates were also compared with those of three reference strains: *B. thuringiensis* IAM 12077 (or ATCC 10792, *B. thuringiensis* type strain); *B. cereus* ATCC 14579 (*B. cereus* type strain); and *B. cereus* F4810/72 (a reference *B. cereus* emetic strain, also known as strain B0358 in the Logan *Bacillus* Collection at Glasgow). Based on the *16S rRNA* fragment, the FC isolates were also closely related to the *B. thuringiensis* and *B. cereus* strains tested.

*B. thuringiensis* is known as one of the closest species to *B. cereus*. These two species are closely related genetically, and their *16S rRNA* sequence identities are more than 99% [29]. Polyphasic identification is therefore useful in this case. As expected from the Blast results, many of the *B. cereus* strains were also among the closest relatives. Because all FC isolates produced parasporal crystals, they were therefore confirmed to be *B. thuringiensis* by their genetic characteristics, which are also supported by the biochemical profiles.

#### 2.2.2. Molecular Typing of Isolates Using Multilocus Sequence Typing (MLST) Analysis

We further examined the genetic diversity and relatedness of the food poisoning isolates using multilocus sequence typing (MLST), which is highly discriminatory for microbial strains. Based on Yang et al. (2017) [30], seven housekeeping genes for the *B. cereus* group (*glpF, gmk, ilvD, pta, pur, pycA, and tpi*) were used in the analysis. A sequence type, ST26, to which *B. cereus* F4810/72 belongs, was also included in the analysis.

Each amplified fragment of the housekeeping genes was sequenced, and the processed sequences were submitted to the MLST database (see Section 4.2.2 for details). The dendrogram constructed using UPGMA with allelic profiles of STs is presented in Figure 2a. The phylogenetic analysis based on sequence alignment revealed that the *B. thuringiensis* isolates belonged to different clonal groups. The isolates FC1, FC6, FC8, FC9, and FC10 were arranged in one clade, and FC2 and FC7 were separated from this clade and were distinct from each other. All of the *B. thuringiensis* isolates were separated from *B. cereus* F4810/72 (Figure 2a).

Using the goeBURST algorithm, one group and two singletons were defined. FC1, FC6, FC7, FC8, FC9, and FC10 (Clonal Complex 8; CC8) were assigned to Group 1, and FC2 and F4810/72 were designated as singletons. Overall, the bacterial strains, including emetic *B. cereus* F4810/72, were resolved into five allelic profiles of STs (Figure 2b), three of which have previously been assigned and two of which are new.

MLST typing analyzes a set of seven housekeeping genes. This method has been successfully used for different applications, including the typing of food-borne isolates [30]. It has also been used for typing of environmental strains of *B. cereus* group [31]. The established *B. cereus* MLST database (PubMLST; http://pubmlst.org/bcereus/info/primers.shtml, accessed on 1 December 2022) allows comparisons of a set of isolates to the existing sequence type and the identification of a new sequence type, under both of which techniques we found our strains belonged.

*B. cereus* F4810/72, a classical emetic strain that was used as a reference in this study, has the allelic profile ST26 [32]. This ST was also the most commonly identified ST for *B. cereus* strains [33]. The fact that the *B. thuringiensis* isolates belonged to different STs clearly confirmed that there was an intraspecific genetic diversity of strains involved in the emetic illness case recorded in this study. Because *B. thuringiensis* was not commonly present in food, and the occurrence of emetic strains with direct evidence of clinical symptoms was even more unusual (Hoffmaster et al., 2008) [33], this finding is worthy of note as it may indicate various sources of contamination or the possibility of gene transfer.

Since *B. thuringiensis* strains have been used as biopesticides, there have been concerns raised about the use of this organism for this purpose due to the potential enterotoxin production of this species. Extensive use of these organisms, especially in the form of spores, may cause *B. thuringiensis* to be more widely spread in the environment and have more frequent implications for food-borne illnesses. Cases of diarrheal disease have been linked to biopesticide strains [34]. Recently, *B. thuringiensis* found in salad was suspected to be the cause of a food-borne outbreak in the EU [29]. Biggel et al. (2022) [34] suggested that many food isolates of *B. thuringiensis* had their origins in biopesticide strains and belong to previously observed sequence types (i.e., ST8, ST15, ST16, or ST23) [35]. Similarly, our MLST results showed that many of the food-borne outbreak *B. thuringiensis* isolates belong to ST8 (four isolates from vomit) and ST15 (one food isolate). These sequence types, which might have been linked to biopesticide strains, have been frequently found in fresh vegetable samples [29,36,37]. Moreover, it is interesting to record the new ST genotypes found in this study and their association with emetic illness. Although we cannot be certain about the original source of the strains belonging to these STs, these data could be useful in future analyses of epidemiological data and assessments of the ecological and biological roles of *B. thuringiensis* isolates. Moreover, the new STs might indicate an emerging pathogenic strain and could be useful in the monitoring of food safety and the control of disease in relation to this rare emetic food-borne pathogen.

### 2.3. Detection and Characterization of Genes Associated with Emetic Traits in B. thuringiensis Isolates

#### 2.3.1. Detection of *Non-Ribosomal Peptide Synthetase* (*NRPS*)/*Cereulide Synthetase* Genes

Genetic factors that are associated with the *Bacillus* emetic traits, i.e., emetic toxin production capability or *cereulide synthetase*, were investigated in the *B. thuringiensis* isolates through PCR amplification of fragments of the *NRPS*/*cereulide synthetase* (*ces*) genes. The expected amplified regions in the *ces* gene cluster (based on the sequence of strain F4810/72, Genbank accession no. DQ360825.1 [38]) are shown in Figure 3a. The PCR results are shown in Figure 3b. Although the amplified gene fragments were achieved with high quantity and specificity with strain F4810/72, the primer pairs had different efficacies when applied to *B. thuringiensis* isolates. The primers BEF/BER, CER1/EMT1, and EM1-F/EM1-R yielded amplicons of expected sizes from all *B. thuringiensis* isolates, even though nonspecific amplifications were also observed in some reactions with primers BEF/BER and EM1F/EM1R. However, primers CesF1/CesR2 gave positive results only with FC2 (food isolate) and FC8 (clinical isolate).

The four pairs of primers used in this investigation were evaluated by different researchers as being specific to *B. cereus* emetic strains or to the *ces* gene. Even though these sets of primers were independently designed, their specific targets were at different sites on the *ces* gene. The BEF/BER and CER1/EMT primers specifically target *cesA*, whereas the EM1F/EM1R and CesF1/CesR2 primers specifically target *cesB* (Figure 3a). Primers BEF/BER, CER1/EMT, and EM1F/EM1R were validated by Toh et al. (2004) [39], Horwood et al. (2004) [40], and Ehling-Schulz et al. (2004) [41], respectively, to be associated with cereulide-producing strains. The CesF1/CesR2 primers were described by Ehling-Schulz et al. (2005b) [42] as specific to the valine activation module of the *cereulide synthetase* gene, and disruption of this gene resulted in a non-cereulide-producing mutant. The differences in the amplification patterns observed in our study for the *B. thuringiensis* isolates and *B. cereus* F4810/72, as seen in Figure 3b, may be due to the variation of the nucleotides within and around these fragments.

Failure to amplify the *ces* fragment by CesF1/CesR2 primers in many *B. thuringiensis* isolates, apart from FC2 and FC8, is not unexpected. Kim et al. (2010) [43] observed that this pair of primers could yield a negative result with some emetic toxin-producing *B. cereus* strains. Our experience showed that this pair of primers required more specific reaction conditions than the others. On the other hand, failure to amplify the gene fragment may be a result of non-conserved DNA sequences around this region of the *cesB* gene in emetic toxin-producing *Bacillus*, especially with sequence diversity within members of the *B. cereus* group having been previously observed [44]. This observation in our study suggested that using these four pairs of primers offers a more sensitive option for identifying the presence of fragments of *cesA* and *cesB,* the two consecutive genes encoding CesA and CesB proteins that are required for cereulide production [45]. Further investigations into the genetic features of the isolates are desirable, and in this study, we explored the sequence of some of these gene fragments.

#### 2.3.2. Sequence Analysis of *B. thuringiensis* Amplified Gene Fragments

Selected gene fragments amplified from the DNA of the *B. thuringiensis* isolates using the primers BEF/BER, CER1/EMT1, and EM1-F/EM1-R were sequenced (the sequences are deposited in the NCBI BioProject database PRJNA188745). They were highly similar (with 98–99% identity) or identical to the *ces* gene found in *B. cereus* F4810/72 (Genbank accession no. DQ360825.1 [38] and the *crs* (another abbreviation for cereulide synthetase) gene from *B. cereus* no. 55 (Genbank accession no. AB248763.2 [46]). Fragments generated using primers CER1/EMT1 were also comparable (with 99–100% identity) to the *B. cereus NRPS* gene (Genbank accession no. AY331260.1 [40]). They also had 99–100% identity with the *cesA* gene found in toxigenic *B. pumilus* NR 19/5 and *B. licheniformis* NR 5106 (Genbank accession no. AM493712.1 and AM493711.1, respectively [47]). In addition, fragments yielded from primers EM1-F/EM1-R also had 96–97% identity to the *cesB* gene found in an emetic toxin-producing strain of *B. pumilus* (strain NIOB133; Genbank accession no. EU289221.1 [48]. The positions on the *B. thuringiensis*’ *ces* gene, which were different from those on the *ces* gene of *B. cereus* and other *Bacillus* species, had mismatches, insertions, or deletions of nucleotides.

From the biochemical and genetic characterization, it is clear that the outbreak isolates do not represent a single strain. The fact that they all harbored fragments that are parts of the *ces* gene cluster indicated that this gene may be distributed among *B. thuringiensis* strains in certain geographical locations [26,49]. Gene transfer is possible given that the *ces* gene in some strains is located on a megaplasmid [44,46,50] and mobile genetic elements (MGEs) have been found to be associated with the *ces* gene cluster [44]. More extensive studies on *ces* distribution in *B. thuringiensis* would help us understand how widely this feature is spread in the natural environment. It would also lead to a better assessment of its impact on food safety.

### 2.4. Cytotoxicity of Emetic B. thuringiensis Isolates to Caco-2 Cells

According to the clinical symptoms and the results from the genetic characterization presented above, we presumed that some of the *B. thuringiensis* isolates would display similar biological activity to that of emetic *B. cereus*, which can produce the emetic toxin cereulide, a heat-stable cyclic peptide [51,52]. Because of the molecular nature of the emetic toxin, which has a very low molecular weight (1.2 KDa), it is difficult to detect it using an immunological method or a simple analytical chemistry method [53]. Therefore, biological assays, especially cytotoxicity assays, have been extensively used for the detection of the emetic toxin [54,55,56].

In this study, the cytotoxicity of the heat-treated culture supernatants of the outbreak isolates was tested on human colorectal adenocarcinoma (Caco-2) cells. The isolates tested included *B. thuringiensis* FC2, FC7, and FC8, which represented sequence types ST2804, 2805, and ST8, respectively. A preliminary cytotoxicity test showed that the heat-treated supernatants of the *B. thuringiensis* isolates were cytotoxic to Caco-2 cells (a non-emetic strain control, *B. cereus* DSM 4384, did not show cytotoxic activity in the same experiment) (Appendix A). Further investigation of the degree of cytotoxicity (expressed as half-maximal cytotoxic concentration, CC_50_) of isolates FC2, FC7, and FC8 showed that the reduction of Caco-2 viability induced by the heat-treated culture supernatants of these isolates was in a dose-dependent manner (Figure 4a). According to the CC_50_ values, the FC2 isolate showed a higher degree of cytotoxicity to Caco-2 cells than FC7 and FC8 (Figure 4a). Vacuolation was observed in Caco-2 cells treated with the heat-treated supernatants from all three isolates (Figure 4b).

It is difficult to identify the isolate(s) responsible for the illness, as the MLST results (Section 2.2.2) indicated a diversity among the outbreak isolates. The cytotoxicity results suggested a possible role of FC2 in the emetic illness, as it was more toxic to Caco-2 cells than the other isolates, although the contribution of the other isolates to the illness may not be excluded.

### 2.5. Annexin V/PI Apoptosis Detection

To clarify the cell death pathway caused by the heat-stable toxin in the supernatants from each bacterial isolate, we performed annexin V/PI staining to discriminate the apoptosis and necrosis mechanisms of cell death through flow cytometry. Caco-2 cells were treated for 48 h with the cell-free supernatants of FC2, FC7, and FC8 at a concentration of 50% (*v*/*v*) (which had 56.62, 62.70, 64.91 µg/mL total protein, respectively). Skim milk and the culture supernatant of F4810/72 were used as negative and positive controls, respectively.

From the results, *B. cereus* F4810/72 appeared to cause both apoptotic cell death (14.85 ± 1.49%) and necrotic cell death (5.91 ± 0.32%) (Figure 5a,b). For cell death through the apoptotic mechanism, 6.07 ± 0.81% was detected as early apoptosis and 8.78 ± 0.69% as late apoptosis. Virtanen et al., (2008) [57] have reported that cereulide from *B. cereus* F4810/72 increased necrotic cell death in porcine fetal Langerhans and beta-cells within 2 days. Similarly, Hoornstra et al. (2013) [58] also observed necrotic cell death caused by cereulide from *B. cereus* F4810/72 in other mammalian cells (PBMC, HaCaT, PK-15, and L-929). Our testing of the effect of heat-stable toxins in the culture supernatant of *B. cereus* F4810/72 with Caco-2 cells also pointed to cell destruction through necrosis. However, apoptosis seemed to be a major mechanism of cell destruction in Caco-2 cells.

As for the *B. thuringiensis* FC2, FC7, and FC8 isolates, the heat-treated cell-free supernatants caused cell death through both the apoptosis and necrosis pathways, as observed with the reference emetic strain *B. cereus* F4810/72. However, necrosis was a major pathway in the FC2 isolate, which had the highest rate of necrosis cell death (23.47 ± 2.92%) among the three *B. thuringiensis* isolates tested (Figure 5b). For FC7 and FC8, cell death rates through the necrosis mechanism were similar to those of apoptosis, which was mainly early apoptosis. These results suggest that the *B. thuringiensis* isolates from this emetic outbreak, although they could cause a similar symptom to that typically recognized as being caused by *B. cereus*, might have some differences in the toxin entity from that of *B. cereus* F4810/72.

## 3. Conclusions

In this study, we report a food-borne emetic illness associated with *B. thuringiensis* and present the results of the investigation on the diversity and pathogenicity of the isolates from this food-borne outbreak incident. The biochemical and genetic profiles suggested an appreciable diversity among the isolates. The multilocus sequence typing (MLST) analysis revealed that the *B. thuringiensis* isolates did not represent one strain. Some isolates belonged to the existing sequence types (ST8 and ST15) that have been linked to biopesticide strains, whereas some belonged to two new STs (designated ST2804 and ST2805). All food and clinical isolates harbored gene fragments located in the *cereulide synthetase* (ces) gene cluster, which indicated the likelihood that an emetic toxin of a similar nature to cereulide might have been implicated in the illness. The heat-stable toxin in the supernatant of *B. thuringiensis* FC2, FC7, and FC8, which represented ST2804, ST2805, and ST8, respectively, reduced the viability of Caco-2 in a dose-dependent manner. According to the CC_50_ values, the FC2 isolate derived from the food involved in the illness showed a higher degree of cytotoxicity to Caco-2 cells than did FC7 and FC8. The flow cytometry with Annexin V/PI staining revealed that cell destruction by all three food-poisoning *B. thuringiensis* isolates occurred through both apoptosis and necrosis pathways, but for FC2, necrosis was likely the main mechanism that caused Caco-2 cell destruction.

## 4. Materials and Methods

### 4.1. Bacterial Strains/Isolates and Identification

The isolates from the food-borne outbreak comprised two from food: FC1 and FC2 (obtained from the leftover pasta sauce: mussels and smoked mackerel in garlic butter sauce) and five from the vomit of one patient: FC6, FC7, FC8, FC9, and FC10. They were isolated on Polymyxin Pyruvate Egg-Yolk Mannitol Bromothymol-Blue Agar (PEMBA), which was incubated at 37 °C for 48 h. The isolates were examined using phase-contrast microscopy and biochemically characterized using the API 20E and API 50 CHB systems (BioMérieux, Marcy-l’Étoile, France). Growth at 40 °C, 42 °C, 45 °C, 48 °C, and 50 °C, was also observed in Tryptone Soy Broth (Oxoid, Basingstoke, UK) cultures incubated in water baths. *B. cereus* F4810/72 (also known as strain B0358 in the Logan *Bacillus* Collection at Glasgow) was used as a reference emetic toxin-producing strain. All bacterial cultures were maintained on Tryptone Soy Agar (TSA) (Oxoid, Basingstoke, UK), stored at 4 °C, or in lyophilized form.

### 4.2. Identification and Typing of B. thuringiensis Isolates from Emetic Food-Borne Illness

#### 4.2.1. Identification of Isolates Using *16S rRNA* Gene Sequencing

Chromosomal DNA from the pure cultures of the emetic food poisoning isolates was extracted using the CTAB/phenol-chloroform extraction method described by Griffiths et al. (2000) [59]. The *16S rRNA* gene was amplified using primers 27F (5′-AGAGTTTGATCMTGGCTCAG-3′) and 1492R (5′-CGGTTACCTTGTTACGACTT-3′) [60] at a final concentration of 0.1 μM. The PCR was performed with an initial denaturation for 2 min at 94 °C, 35 cycles of denaturation for 30 s at 94 °C, primer annealing for 30 s at 55 °C, extension for 30 s at 72 °C, and a final extension for 7 min at 72 °C (Lane, 1991) [61]. The PCR products were purified using a PCR purification kit (Sigma-Aldrich, Darmstadt, Germany). Sequencing was performed by Celemics (Seoul, Republic of Korea). The low-quality bases in the sequences were trimmed using Bioedit software version 7.05.3 [62]. The processed sequences were subjected to Basic Local Alignment Search Tool (BLAST) analysis against the *16S rRNA* sequences in the National Center for Biotechnology Information (NCBI) database to find the closest relatives.

#### 4.2.2. Typing of Isolates Using Multilocus Sequence Typing (MLST) Analysis

The primers for the housekeeping genes (*glpF*, *gmk*, *ilvD*, *pta*, *pur*, *pycA*, and *tpi*) were used for multilocus sequence typing (MLST) according to Yang, et al. (2017) [30] *(ilvD2* was also used if the *ilvD* locus failed to find an allelic profile). The sequencing of the amplified genes was performed by Macrogen (Seoul, Republic of Korea).

Sequences were analyzed using BioEdit software version 7.1.9 (Abbott, Carlsbad, CA, USA). All allelic profiles and sequence types (STs) detected in the study were assigned according to the online MLST database for *B. cereus* (https://pubmlst.org/organisms/ bacillus-cereus, accessed on 1 December 2022). Grouping tools were used to establish the molecular type, clonal complexes (CCs), singletons, and the relationship between established groups. All STs were displayed with BURST [63], using the characteristic profiles for each isolate as the input data. Minimal expansion trees were also generated using the geoBURST algorithm [63] performed with Phyloviz 2.0 [64] to define CCs and display models. A phylogenetic tree of the related sequences of the seven loci of the housekeeping genes was generated with UPGMA based on the method of Kimura-2-parameter [65] and performed with 1000 bootstrap replicates in Mega X [66].

### 4.3. Genetic Analysis of B. thuringiensis Isolates

#### 4.3.1. DNA Extraction

DNA samples were prepared from overnight cultures of *Bacillus* strains grown on Trypticase Soy Agar (TSA) slants (Oxoid, Basingstoke, UK). Cells were scraped from the surface of the agar slant into TE buffer (10 mM Tris, 1 mM EDTA, pH 8.0), and the DNA was extracted using a CTAB-based method for mini-preparation of DNA [67].

#### 4.3.2. Polymerase Chain Reaction (PCR) Amplification of Cereulide Synthetase Gene

Amplification of the *cereulide synthetase* (*ces*) gene was carried out using four pairs of oligonucleotide primers: BEF/BER [39], CER1/EMT1 [40], EM1-F/EM1-R [41], and CesF1/CesR2 [42]. The annealing temperatures for the above primers were 55 °C, 58 °C, 61 °C, and 58 °C, respectively (optimized in this study for amplification of these fragments on *B. thuringiensis* DNA). The *internal transcribed spacer* (*its*) gene was used for an internal control, which was amplified using primers ITS-F1/ITS-R1 [68] with an annealing temperature of 50 °C. Each of the 50 mL-reactions contained 1 × reaction buffer, 1.5 mM MgCl_2_, 0.2 mM dNTPs, 20 pmoles of each primer, 1 μL DNA template, and 1 unit of DNA polymerase (Phusion High-Fidelity DNA Polymerase, New England Biolab, Ipswich, MA, USA). Amplification was carried out in the PTC 200 thermal cycler (MJ Research, Waltham, MA, USA), using the following steps: 5 min initial denaturation at 95 °C; 35 cycles of PCR amplification for 45 s at 95 °C, 45 s at 50–61 °C (depending on the primers, see above), 1.5 min at 72 °C; and 10 min final extension at 72 °C.

#### 4.3.3. DNA Sequencing and Sequence Similarity Analysis

Purified PCR products from selected *B. thuringiensis* isolates amplified using primers CER1/EMT1, BEF/BER, and EM1-F/EM1-R were sequenced using an Applied Biosystems genetic analyzer (performed by First Base Laboratory, Selangor, Malaysia). The nucleotide sequences of the PCR fragments were analyzed for their similarities to sequences published in the NCBI database using the Blast program. In addition, they were compared to the sequence of reference strains using the Blast 2 Sequences program.

### 4.4. Cytotoxicity Assay

#### 4.4.1. Preparation of Heat-Treated Supernatants from Bacterial Isolates

The overnight bacterial cultures (0.1% (*v/v*)) were inoculated in 50 mL of 2.5% skim milk medium (SMM) (Oxoid, Basingstoke, UK). The cultures were incubated for 24 h at 25 °C in an orbital shaking incubator (shaker speed: 150 rpm). After incubation, the supernatant portion of each bacterial culture was collected after centrifugation at 3000× *g* for 10 min, and the cell pellet was discarded. The supernatant was heated in an autoclave for 15 min at 121 °C, left to cool, and filtered through a 0.2 µm membrane filter [56].

#### 4.4.2. Cell Culture

The human colorectal adenocarcinoma cell line (Caco-2) (ATCC HTB-37) was cultured in Dulbecco’s Modified Eagle Medium (DMEM; Gibco; Thermo Fisher Scientific, Waltham, MA, USA) was supplemented with 10% heat-inactivated fetal bovine serum (Gibco; Thermo Fisher Scientific, Waltham, MA, USA) and Pen–Strep (100 units/mL penicillin and 100 μg/mL streptomycin) (Caisson, Smithfield, UT, USA). The cells were incubated at 37 °C in a 5% CO_2_ incubator.

#### 4.4.3. Cytotoxicity Assay

The cytotoxicity of the heat-treated supernatant from the bacterial isolates was tested using the MTT cytotoxicity assay [56] and the PrestoBlue cytotoxicity assay [69]. In brief, Caco-2 cells were plated at 10^5^ cells/well in 96-well plates for 48 h before the experiment. The Caco-2 cells were treated with 100 µL of the supernatant from the bacterial isolates prepared at various concentrations for 48 h. For the PrestoBlue cytotoxicity test, after 48 h of treatment, the PrestoBlue™ reagent (Thermo Fisher Scientific, Waltham, MA, USA) was added to measure cell viability, and the cells were incubated for 30 min at 37 °C in a 5% CO_2_ incubator. Absorbance was read at 570/600 nm using a Bio Tek Synergy HTX microplate reader (Agilent Technologies, Santa Clara, CA, USA). The percentages of cell viability in relation to that of the non-treatment control were calculated using the following equation.
% Cell viability = [(OD_570_ − OD_600_) treated cells/(OD_570_ − OD_600_) non-treated cells] × 100(1)

Half-maximal cytotoxic concentration (CC_50_) values were then calculated using Graph Pad Prism 9.4.1 software (GraphPad Software, Inc., San Diego, CA, USA). Protein concentrations of the heat-treated supernatants were measured using Bradford reagent (Thermo Fisher Scientific, Waltham, MA, USA) according to the manufacturer’s protocol.

#### 4.4.4. Annexin V/PI Assay

Apoptotic stages were evaluated by conducting an Annexin V/PI assay and analyzed using a flow cytometer. Caco-2 cells were seeded at 5 × 10^4^ cells/mL in 6-well culture plates, using 2 mL per well, and treated with the supernatant from the bacterial isolates at 50% concentration or 1/2 dilution (having total proteins of 56.62 μg/mL, 62.70 μg/mL, 64.91 μg/mL for FC2, FC7, and FC8, respectively). After 48 h of treatment, the Caco-2 cells were collected and stained with Annexin V/PI, according to the Annexin V/PI kit protocol (ImmunoTool, Friesoythe, Germany). Briefly, the cell pellets were trypsinized and resuspended in 100 µL of ice-cold 1 × binding buffer. Annexin V-APC and PI were added to the tubes at the dilution of 1:100. The cell suspension was then mixed and incubated for 15 min in the dark. Samples were analyzed on a BD Accuri C6 Flow Cytometer and FlowJo software (BD Biosciences, San Jose, CA, USA).

## Figures and Tables

**Figure 1 toxins-15-00089-f001:**
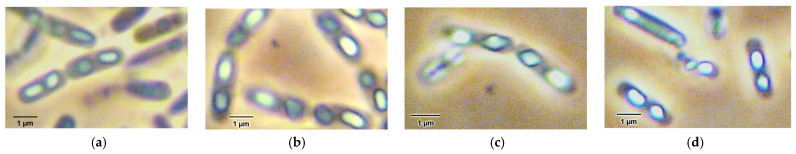
Phase-contrast micrographs of some *B. thuringiensis* isolates obtained from food and clinical (vomit) samples associated with the emetic illness: (**a**) FC1 (food isolate); (**b**) FC2 (food isolate); (**c**) FC7 (clinical isolate); (**d**) FC8 (clinical isolate). The pictures show cells containing ellipsoidal spores and bipyramidal toxin crystals.

**Figure 2 toxins-15-00089-f002:**
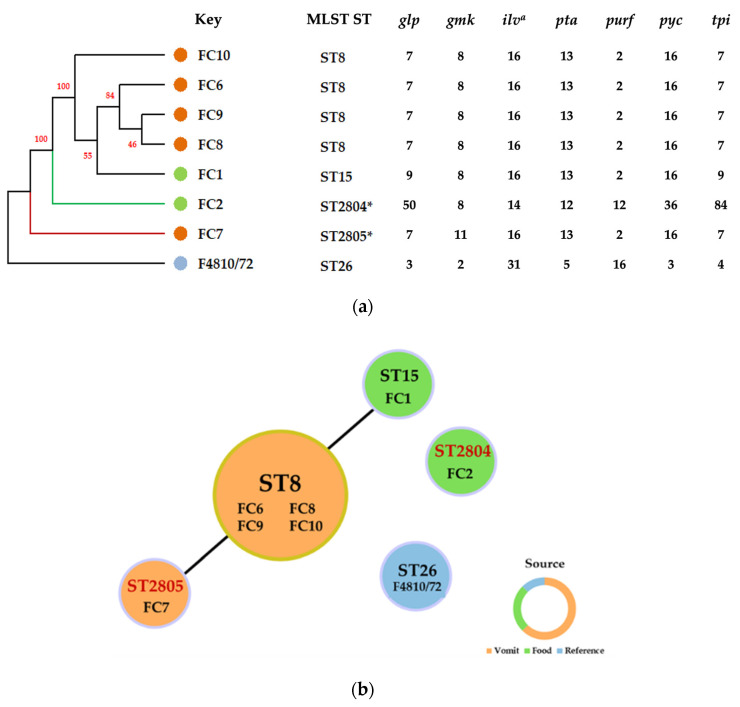
Multilocus sequence typing (MLST) analysis of the *Bacillus thuringiensis* group isolates: (**a**) Comparison of UPGMA-based dendrograms of eight isolates and the MLST profiles. The two new MLST profiles were submitted to the PubMLST database and are marked with an asterisk (*). (**b**) goeBURST analysis performed with Phyloviz2.0, representing the clustering of the *B. thuringiensis* food poisoning isolates and one reference emetic *B. cereus* strain into 5 sequence types (STs). The ST identification, according to the MLST database, and the isolate/strain belonging to each ST are indicated inside each circle. The circle size is proportional to the number of isolates belonging to the same ST type. The new STs are shown in red.

**Figure 3 toxins-15-00089-f003:**
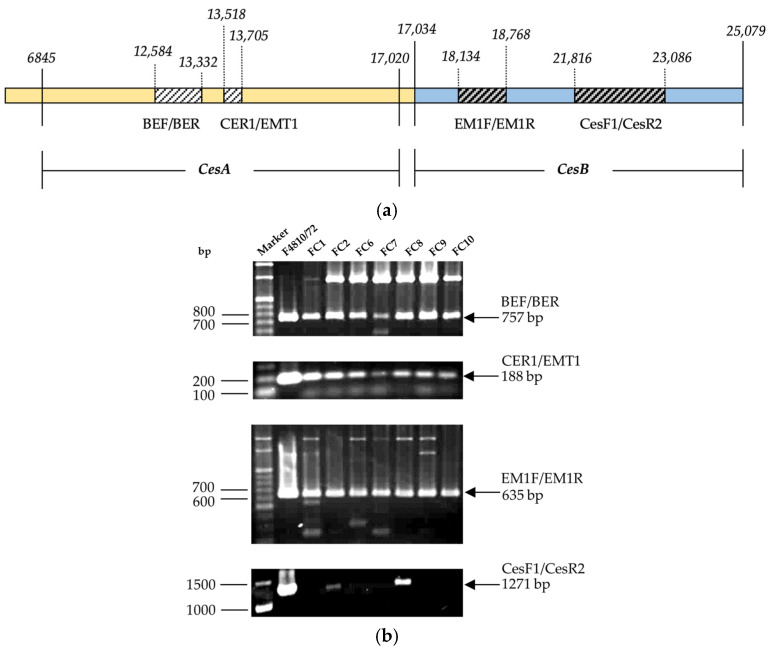
(**a**) A diagram illustrating the relative positions of the target sites of primers BEF/BER, CER1/EMT1, EM1F/EM1R, and CesF1/CesR2 on the *cesA* and *cesB* genes. The positions shown in the diagram are based on the sequence derived from *B. cereus* strain F4810/72 (Genbank accession no. DQ360825.1). (**b**) Specific amplicons (indicated by arrows) obtained from PCR using different primer sets.

**Figure 4 toxins-15-00089-f004:**
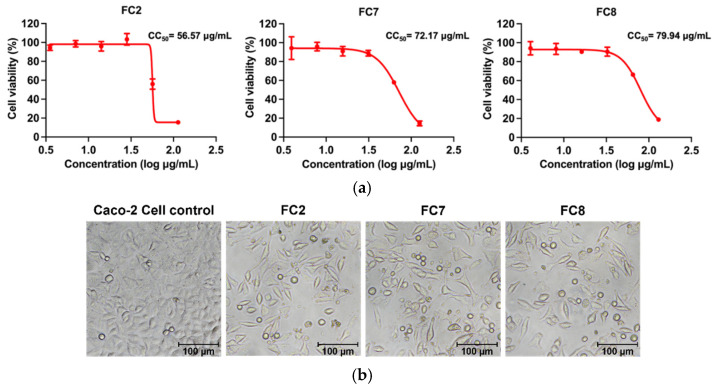
Concentration-response curves for heat-stable toxin in the culture supernatants of the *B. thuringiensis* isolates (FC2, FC7, and FC8) involved in the emetic food-borne outbreak (**a**) and the effect of the heat-treated culture supernatants of the isolates (50% (*v/v*)) on Caco-2 cells (**b**). The data are presented with SD (*n* = 3). The curves were generated with Graph Pad Prism 9.4.1 using a nonlinear regression curve-fitting algorithm.

**Figure 5 toxins-15-00089-f005:**
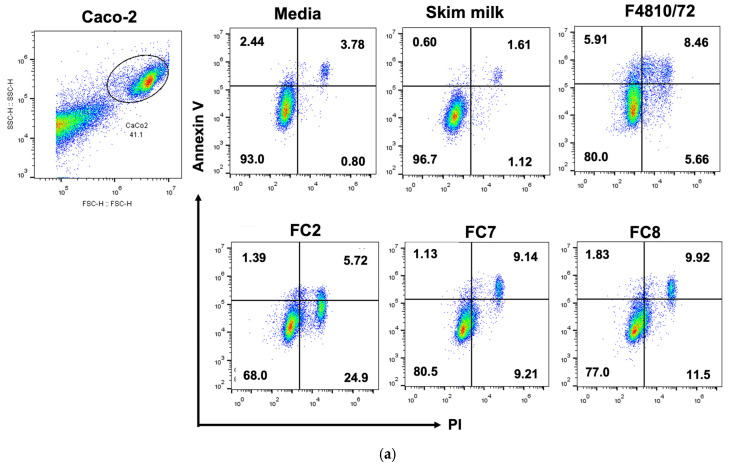
Results from flow cytometric analysis of Caco-2 cells treated with culture supernatants of *B. cereus* F4810/72 (B0358) and *B. thuringiensis* FC2, FC7, and FC8 isolates, showing that the strains induced apoptosis and necrosis cell death in Caco-2. The cells were treated with 50% (*v/v*) of the culture supernatants for 48 h before staining with annexin V/PI. (**a**) Four distinct cell phenotypes are distinguishable by the quadrant: viable (lower left quadrant), early apoptosis (upper left quadrant), late apoptosis (upper right quadrant), and necrosis (lower right quadrant). (**b**) The bar graphs show the percentages (as means ± SD) of cells treated with the culture supernatants from the test strains in the early apoptosis, late apoptosis, and necrosis phases, which were statistically significantly different (***, *p* < 0.005) from the control (untreated cells).

**Table 1 toxins-15-00089-t001:** Closest strains to the FC isolates involved in the emetic food-borne illness, retrieved from the Blast results of their *16S rRNA* sequences against strains in the NCBI nucleotide database.

Isolate	Accession No. (FC Isolate)	Closest Relative	Accession No. (Closest Relative)	Percent Identity	Percent Coverage
FC1	ON351569	*Bacillus thuringiensis* HER1410	CP050183.1	100.00	93
FC2	ON351570	*Bacillus thuringiensis* GCU1	MN590524.1	99.93	97
		*Bacillus thuringiensis* 41(7) il	KY323326.1	100.00	97
FC6	ON351571	*Bacillus thuringiensis* HER1410	CP050183.1	99.93	99
FC7	ON351572	*Bacillus thuringiensis* HER1410	CP050183.1	100.00	98
FC8	ON351573	*Bacillus thuringiensis* HER1410	CP050183.1	100.00	99
FC9	ON351574	*Bacillus thuringiensis* HER1410	CP050183.1	100.00	97
FC10	ON351575	*Bacillus thuringiensis* LAA3	OM585510.1	100.00	95

## Data Availability

Not applicable.

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
