# Peer review of "Intraspecific Diversity and Pathogenicity of Bacillus thuringiensis Isolates from an Emetic Illness"

_toxins, 2023, doi:10.3390/toxins15020089_

Round 1

Reviewer 1 Report

The study is interesting and well detailed.

Point 2.2.3 It's not clear the grade of relatedness among the different isolates. The authors's choice to use three different methods to analyse and to graphically represent the relations is redundant and confusing. It would be better to choose one approach and to describe it in detail. Is there a reason why whole genome sequencing was not considered?

Reviewer 2 Report

After reading the manuscript entitled "Intraspecific Diversity and Pathogenicity of Bacillus thuringiensis Isolates from an Emetic Illness", I would like to make a few comments:

1) It seems to me that the "Introduction" section could be a little more detailed; the authors could describe in more detail the clinical picture and epidemiology of foodborne illnesses caused by the B. cereus complex.

2) The authors are conducting some kind of epidemiological investigation - they are studying strains isolated from clinical samples and from a potential source of infection. But I did not find an answer to the question  -  which strain caused the disease?  Was it a particular strain, or all of them, or a specific combination of them? If it was several strains, how could several pathogenic strains infect a person at the same time?

I understand that it is difficult to establish this unambiguously, but the authors could at least discuss these issues in the text.

3) In my opinion there are some flaws in the methodology for assessing the toxicity of supernatants. If the task is to investigate the activity of short peptides, then larger proteins should be excluded from the studied supernatants, because these proteins can also have cytotoxic activity. In addition, there are no controls in Figure 6A - at least the survival curve of cells treated with the supernatant of a strain lacking ces genes

Round 2

Reviewer 2 Report

In my opinion, the authors have significantly improved their manuscript. The revised version takes into account all my comments.Nevertheless I would like to make one minor remark:

Lines 143-144

MLST typing analyzes a set of seven housekeeping genes, therefore, it is a high-resolution typing method.

I do not fully agree with this formulation. The resolving power of MLST is much lower than that of other typing methods.

Author Response

Response to Reviewer 2

Comments: In my opinion, the authors have significantly improved their manuscript. The revised version takes into account all my comments.Nevertheless I would like to make one minor remark:

Lines 143-144

MLST typing analyzes a set of seven housekeeping genes, therefore, it is a high-resolution typing method.

I do not fully agree with this formulation. The resolving power of MLST is much lower than that of other typing methods.

Response: Thank you. We have changed the sentence, now it is written “MLST typing analyzes a set of seven housekeeping genes.” (Line 163 in the revised manuscript, revision 2).

Thank you very much for your comments that help us to improve our manuscript. They are very much appreciated.